

# Prognostic value of KRAS mutation status in colorectal cancer patients: a population-based competing risk analysis

Dongjun Dai[1], Yanmei Wang[1], Liyuan Zhu[2], Hongchuan Jin[2] and Xian Wang[1]

[1] Department of Medical Oncology, Sir Run Run Shaw Hospital, Medical School of Zhejiang University, Zhejiang University, Hangzhou, Zhejiang, China
[2] Laboratory of Cancer Biology, Key Lab of Biotherapy, Sir Run Run Shaw Hospital, Medical School of Zhejiang University, Zhejiang University, Hangzhou, Zhejiang, China

Corresponding author
Xian Wang, wangx118@zju.edu.cn

## ABSTRACT

**Background:** To use competing analyses to estimate the prognostic value of KRAS mutation status in colorectal cancer (CRC) patients and to build nomogram for CRC patients who had KRAS testing.

**Method:** The cohort was selected from the Surveillance, Epidemiology, and End Results database. Cumulative incidence function model and multivariate Fine-Gray regression for proportional hazards modeling of the subdistribution hazard (SH) model were used to estimate the prognosis. An SH model based nomogram was built after a variable selection process. The validation of the nomogram was conducted by discrimination and calibration with 1,000 bootstraps.

**Results:** We included 8,983 CRC patients who had KRAS testing. SH model found that KRAS mutant patients had worse CSS than KRAS wild type patients in overall cohort (HR = 1.10 (95% CI [1.04–1.17]), $p < 0.05$), and in subgroups that comprised stage III CRC (HR = 1.28 (95% CI [1.09–1.49]), $p < 0.05$) and stage IV CRC (HR = 1.14 (95% CI [1.06–1.23]), $p < 0.05$), left side colon cancer (HR = 1.28 (95% CI [1.15–1.42]), $p < 0.05$) and rectal cancer (HR = 1.23 (95% CI [1.07–1.43]), $p < 0.05$). We built the SH model based nomogram, which showed good accuracy by internal validation of discrimination and calibration. Calibration curves represented good agreement between the nomogram predicted CRC caused death and actual observed CRC caused death. The time dependent area under the curve of receiver operating characteristic curves (AUC) was over 0.75 for the nomogram.

**Conclusion:** This is the first population based competing risk study on the association between KRAS mutation status and the CRC prognosis. The mutation of KRAS indicated a poor prognosis of CRC patients. The current competing risk nomogram would help physicians to predict cancer specific death of CRC patients who had KRAS testing.

## INTRODUCTION

Colorectal cancer (CRC) is the second and third most common cancer of women and men worldwide, respectively (*Bray et al., 2018*). The amount of deaths due to CRC ranked the

second among all cancer types in 2018 (*Bray et al., 2018*). CRC is a heterogeneous disease with various genetic events (*Inamura, 2018*; *Punt, Koopman & Vermeulen, 2017*). Target therapy such as anti-epidermal growth factor receptor (EGFR) therapy has been developed for metastatic CRC (*Chan et al., 2017*).

KRAS is an effector molecule that makes the signal transduction from ligand-bound EGFR to the nucleus (*Liu, Wang & Li, 2019*). KRAS has intrinsic GTPase activity and it binds to GTP to active downstream pathway, such as RAS/RAF/MAPK and PI3K/AKT pathways, to promote cell proliferation. Normally, the GTPase activating proteins would enhance the GTPase activity of KRAS and transform the status of GTP-bound KRAS into a status of GDP-bound KRAS, terminating the downstream signaling. However, some types of KRAS mutation could impair the GAP binding to KRAS and lead to a continuous GTP-bound KRAS status to promote the proliferation related pathways and cancer development (*Cox & Der, 2010*). The mutation of KRAS would also impair the efficacy of EGFR-targeted therapy (*Liu, Wang & Li, 2019*). KRAS mutation is found in about 33–45% of CRC (*Tan & Du, 2012*). Hence, the KRAS testing is recommended for CRC patients who would receive anti-EGFR therapy. The anti-EGFR therapy is limited to KRAS wild type (WT) CRC patients but not KRAS mutant (MT) patients (*Markman et al., 2010*).

Despite the KRAS mutation status as a biomarker for the anti-EGFR therapy of CRC patients, whether it is an independent prognostic factor in CRC was controversial. In metastatic CRC, there were studies showed that KRAS MT patients had worse progression-free survival (PFS) (*Modest et al., 2016*; *Souglakos et al., 2009*) or overall survival (OS) (*Modest et al., 2016*) than KRAS WT patients, while other study found there was no association between KRAS mutation status and PFS or OS of CRC patients (*Kim et al., 2016*). Among stage II and III CRC, there were studies found KRAS mutation would worsen the OS (*Richman et al., 2009*) or disease-free survival (DFS) (*Deng et al., 2015*; *Lee et al., 2015*) of patients while other study found KRAS mutation was not associated with the OS or recurrence-free survival (RFS) of CRC patients (*Roth et al., 2010*). In stage III colon cancer, a study found KRAS mutation status was not associated with the OS or RFS or DFS of patients (*Ogino et al., 2009*), while more recently studies found the KRAS mutation would worsen the DFS (*Sperlich et al., 2018*) or survival after recurrence (SAR) (*Taieb et al., 2019*) of patients. To be noted, most of these studies included with limited amount of CRC patients who had KRAS testing.

The Surveillance, Epidemiology, and End Results (SEER) database of the National Cancer Institute is a national collaboration program of United States, covering 34.6% of the national population. It collects the incidence, survival and treatment data of cancer patients. There was a SEER based study (*Charlton et al., 2017*) on the association between KRAS mutation status and the OS of patients with left or right side CRC. However, despited that CRC is an aggressive disease, the median age at diagnosis for colon cancer patients is 68 years in men and 72 years in women, respectively; for rectal cancer patients it is 63 years in both men and women (*Society, 2017*). In this case, competing risk events might be involved, as the elders might die from diseases other than CRC such as cardiovascular disease (*Zhang, 2017*). Competing risk models such as the cumulative

**Table 1 The characteristic of each included variables in KRAS MT and KRAS WT patients.**

| Characteristics | KRAS MT | | KRAS WT | | p Value |
|---|---|---|---|---|---|
| | No. of patients | % | No. of patients | % | |
| Age | | | | | **0.045** |
| <29 | 38 | 1.05 | 70 | 1.30 | |
| 30–39 | 145 | 4.01 | 246 | 4.58 | |
| 40–49 | 503 | 13.91 | 738 | 13.75 | |
| 50–59 | 820 | 22.68 | 1,261 | 23.50 | |
| 60–69 | 992 | 27.43 | 1,435 | 26.74 | |
| 70–79 | 749 | 20.71 | 995 | 18.54 | |
| >=80 | 369 | 10.20 | 622 | 11.59 | |
| Sex | | | | | **0.023** |
| Female | 1,697 | 46.93 | 2,387 | 44.48 | |
| Male | 1,919 | 53.07 | 2,980 | 55.52 | |
| Race | | | | | **<0.001** |
| White | 2,757 | 76.24 | 4,264 | 79.45 | |
| African Americans | 518 | 14.33 | 598 | 11.14 | |
| Others | 328 | 9.07 | 490 | 9.13 | |
| Unknown | 13 | 0.36 | 15 | 0.28 | |
| Location | | | | | **<0.001** |
| Left | 1,208 | 33.41 | 2,268 | 42.26 | |
| NOS | 97 | 2.68 | 160 | 2.98 | |
| Rectum | 665 | 18.39 | 992 | 18.48 | |
| right | 1,646 | 45.52 | 1,947 | 36.28 | |
| Tumor size | | | | | 0.086 |
| <=2 cm | 399 | 11.03 | 647 | 12.06 | |
| >6 | 674 | 18.64 | 997 | 18.58 | |
| 2–4 | 855 | 23.64 | 1,363 | 25.40 | |
| 4–6 | 1,054 | 29.15 | 1,476 | 27.50 | |
| N | 634 | 17.53 | 884 | 16.47 | |
| Surgery | | | | | **0.005** |
| No | 816 | 22.57 | 1,079 | 20.10 | |
| Yes | 2,800 | 77.43 | 4,288 | 79.90 | |
| Stage | | | | | **<0.001** |
| 0/(Tis) | 13 | 0.36 | 12 | 0.22 | |
| I | 202 | 5.59 | 384 | 7.15 | |
| II | 461 | 12.75 | 817 | 15.22 | |
| III | 887 | 24.53 | 1,494 | 27.84 | |
| IV | 2,020 | 55.86 | 2,615 | 48.72 | |
| Unknown | 33 | 0.91 | 45 | 0.84 | |
| Grade | | | | | **<0.001** |
| Low grade (I & II) | 2,502 | 69.19 | 3,490 | 65.03 | |
| High grade (III & IV) | 712 | 19.69 | 1,370 | 25.53 | |

(Continued)

| Characteristics | KRAS MT | | KRAS WT | | p Value |
|---|---|---|---|---|---|
| | No. of patients | % | No. of patients | % | |
| NOS | 402 | 11.12 | 507 | 9.45 | |
| Regional nodes positive | | | | | **0.025** |
| >=10 | 1,196 | 33.08 | 1,726 | 32.16 | |
| 0 | 903 | 24.97 | 1,485 | 27.67 | |
| 1–3 | 880 | 24.34 | 1,212 | 22.58 | |
| 4–9 | 637 | 17.62 | 944 | 17.59 | |
| Radiotherapy | | | | | 0.383 |
| No | 3,058 | 84.57 | 4,576 | 85.26 | |
| Yes | 535 | 14.80 | 772 | 14.38 | |
| Unknown | 23 | 0.64 | 19 | 0.35 | |
| Chemotherapy | | | | | **0.002** |
| No | 924 | 25.55 | 1,534 | 28.58 | |
| Yes | 2,692 | 74.45 | 3,833 | 71.42 | |
| Marital status | | | | | 0.237 |
| Married | 1,988 | 54.98 | 2,958 | 55.11 | |
| Unmarried | 1,464 | 40.49 | 2,204 | 41.07 | |
| Unknown | 164 | 4.54 | 205 | 3.82 | |

Note:

KRAS MT, KRAS mutant; KRAS WT, KRAS wild type; the widowed or single (never married or having a domestic partner) or divorced or separated patients was defined as unmarried; Tis, Tumor in situ; p value referred to the difference between MT and WT KRAS patients; the significant p values were bolded.

incidence function (CIF) model and Fine-Gray regression for proportional hazards modeling of the subdistribution hazard (SH) model (*Austin, Lee & Fine, 2016*) should be used for the prognostic analyses of population based studies of CRC.

A nomogram is a useful method to predict the probability of patients' clinical outcomes (*Balachandran et al., 2015*). It has compared favorably to traditional TNM staging systems in the prognostic prediction in a series of cancers (*Bobdey et al., 2018*; *He et al., 2018*). To our knowledge, there is currently no nomogram constructed for predicting the outcomes of CRC patients who had KRAS testing.

Here we performed a SEER based study to evaluate the association between KRAS mutation status and the cancer specific survival (CSS) of CRC patients by using competing risk analyses. We also drew an SH model based nomogram for the cancer specific death prediction of CRC patients who had KRAS testing.

## METHODS

### Cohort information

The SEER based cohort was selected using SEER*Stat 8.3.5 software (SEER ID: daid). The access to Collaborative Stage Site-Specific Factor 9 (KRAS mutation status) was granted by the National Cancer Institute (NCI). We included patients who met the inclusion criteria as the follows: (1) it should be a CRC patient who had KRAS testing; (2) it

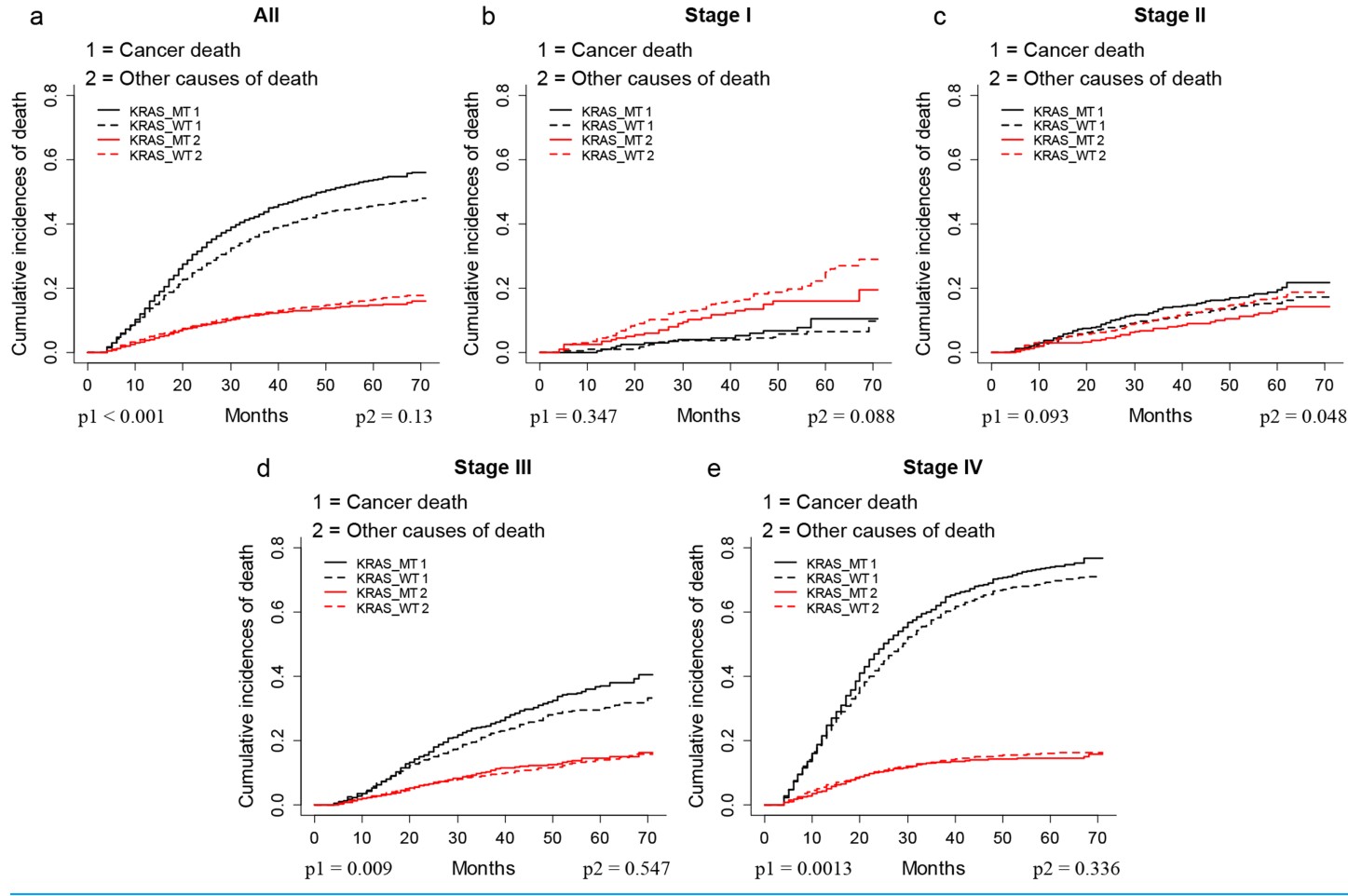

**Figure 1** **CSS of CRC patients with different stages according to KRAS status by CIF plot.** CIF plots of KRAS status and the prognosis of CRC in overall population (A) and stage I–IV CRC patients (B–E).

should include sufficient clinicopathological information of the variables in current study (Table 1). As the information of KRAS testing was collected since 2010, we only included patients who were diagnosed equal to or after 2010. Finally, as shown in Fig. S1, to find an adequate follow-up time, the patients diagnosed between 2010 and 2012 were included. For tumor location, left side means the tumors in splenic flexure, descending colon, sigmoid and rectosigmoid junction, and right side means the tumors in cecum, ascending colon, hepatic flexure and transverse. We defined the median follow-up as the median observed survival time. The last follow-up time was December 31, 2015.

## Statistical analyses

The chi-square test was applied for the comparisons of difference variables between KRAS WT and KRAS MT CRC patients. The cumulative incidences of death (CID) was estimated for cancer related deaths and non-cancer related deaths. Multivariate SH model, which involved all variables, was used to assess the CSS of CRC patients. SH model based nomogram was constructed to predict the 1-year, 2-year and 3-year CSS of CRC patients who had KRAS testing. To be noted, many prediction factors in one model might

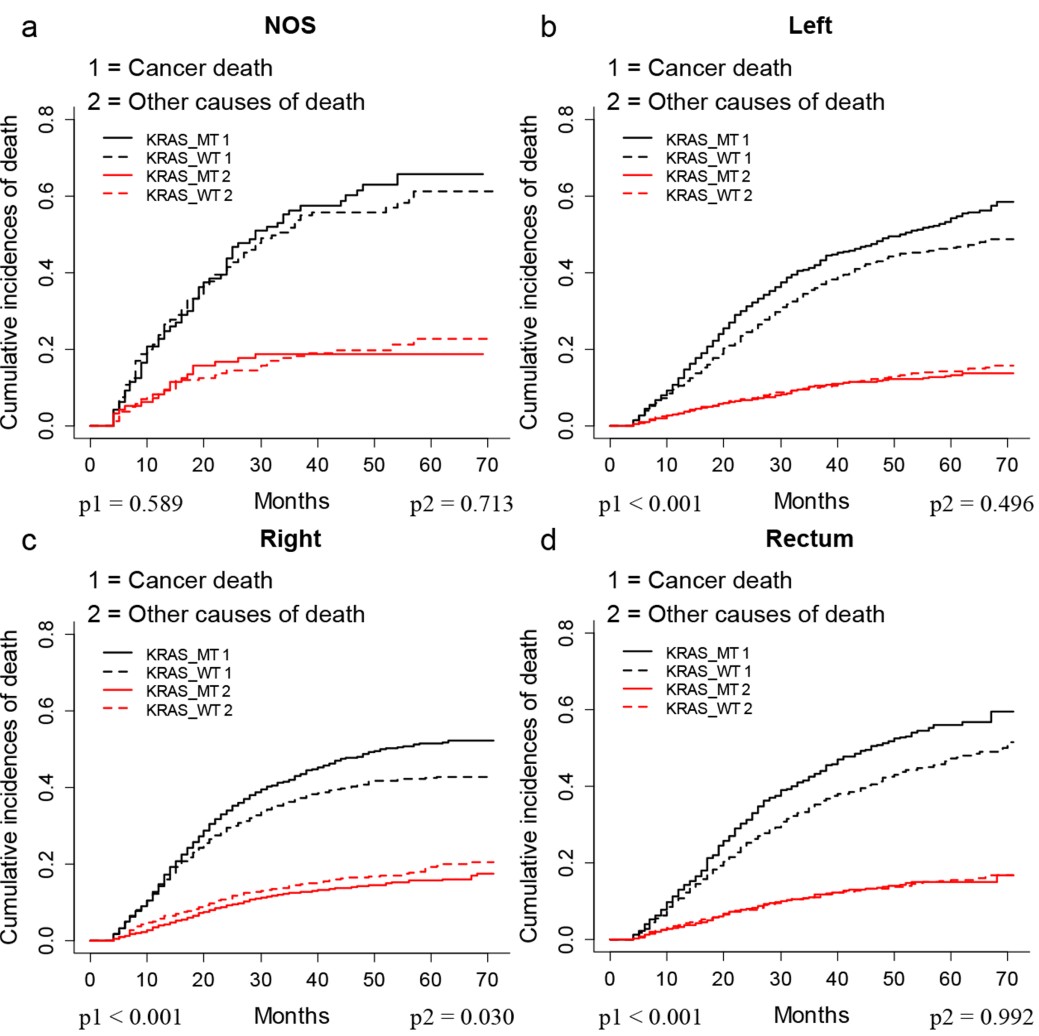

**Figure 2 CSS of CRC patients with differed location according to KRAS status by CIF plot.** CIF plots of KRAS status and the prognosis of CRC in locations of unknown (A), left colon (B), right colon (C) and rectum (D) NOS, not otherwise specified.

cause over-fitting. Hence, we used the variable selection to improve the interpretation and the accuracy of prediction of the competing nomogram (*Ha et al., 2014*). Penalized variable selection was performed by using methods of least absolute shrinkage and selection operator (LASSO), measure–correlate–predict (MCP) and smoothly clipped absolute deviation (SCAD) to select variables for SH model based nomogram. This nomogram was internally validated by discrimination and calibration with 1,000 times bootstraps (*Balachandran et al., 2015*). The calibration curves and the area under the curve of receiver operating characteristic curve (AUC) were used for discrimination and calibration, respectively.

The statistical analyses of current study were performed by a series of packages in R version 3.5.1. The detailed using of those packages could be found in our previous published study (*Dai et al., 2020*). We considered a *p*-value less than 0.05 as statistically significant.

**Table 2 Subgroup analysis of KRAS mutation status and the prognosis of CRC patients.**

| Group | Patients amount | KRAS MT vs. KRAS WT Multivariate SH model HR (95% CI) |
|---|---|---|
| All | 8,983 | **1.10 [1.04–1.17]** |
| Stage I | 586 | 1.30 [0.62–2.70] |
| Stage II | 1,278 | 1.27 [0.95–1.69] |
| Stage III | 2,381 | **1.28 [1.09–1.49]** |
| Stage IV | 4,635 | **1.14 [1.06–1.23]** |
| Unknown location | 257 | 1.01 [0.69–1.49] |
| Left | 3,476 | **1.28 [1.15–1.42]** |
| Right | 3,593 | 1.07 [0.97–1.19] |
| Rectum | 1,657 | **1.23 [1.07–1.43]** |

Note:
KRAS MT, KRAS mutant; KRAS WT, KRAS wild type; HR, Hazard ratios; 95% CI, 95% confidence intervals; the significant results were bolded.

## RESULTS

### Cohort information

As shown in Table 1, there were totally 8,983 CRC patients (3,616 KRAS MT patients and 5,367 KRAS WT patients) included in current study. Significant differences were found between KRAS MT and KRAS WT patients among variables of age, race, location, surgery, tumor stage, grade, positive regional nodes amount, and chemotherapy experience ($p < 0.05$). In detail, compared with KRAS WT patients, the KRAS MT patients had more African American race (14.33% vs. 11.14%), more occurrence in right side of the colon (45.52% vs. 36.28%), less surgery performance (77.43% vs. 79.90%), more metastatic site (55.86% vs. 48.72%), lower grade (grade III & IV: 19.69% vs. 25.53%), and more chemotherapy experience (74.45% vs. 71.42%). The median follow-time were 30 months and 36 months for KRAS MT and KRAS WT, respectively. In KRAS MT patients, the death rate caused by cancer and other reasons were 49.89% and 13.69%, respectively. In KRAS WT patients, the death rate caused by cancer and other reasons were 42.59% and 14.83%, respectively.

### KRAS MT patients had worse outcomes than KRAS WT patients

The CIF plots showed that the KRAS MT patients had a worse CSS than KRAS WT patients ($p < 0.001$, Fig. 1A). We further performed subgroup analysis of KRAS mutation status among different AJCC 7th stages and tumor locations. The CIF plots found that KRAS mutation had no association with the CSS of stage I ($p = 0.347$, Fig. 1B) and stage II ($p = 0.093$, Fig. 1C) CRC patients while it contributed to worse CSS in stage III ($p = 0.009$, Fig. 1D) and stage IV ($p = 0.0013$, Fig. 1E) CRC patients. In addition, the CIF plots showed that KRAS mutation was a hazard factor for the CSS of patients with cancers in the location of left colon, right colon and rectum ($p < 0.001$, Fig. 2).

As shown in Table 2, the multivariate SH model showed that KRAS MT patients had worse CSS (Hazard ratio (HR) = 1.10, 95% CI (95% confidence index) = 1.04–1.17,

**Table 3 Multivariate SH analyses of each variables in KRAS MT and WT patients.**

| Characteristics | SH model | |
|---|---|---|
| | KRAS MT<br>HR (95% CI) | KRAS WT<br>HR (95% CI) |
| Age | | |
| <29 | Reference | Reference |
| 30–39 | 0.58 [0.36–0.94] | 1.19 [0.85–1.66] |
| 40–49 | 0.71 [0.46–1.11] | 1.05 [0.77–1.44] |
| 50–59 | 0.72 [0.46–1.13] | 1.11 [0.82–1.51] |
| 60–69 | **0.59 [0.38–0.92]** | 1.03 [0.76–1.40] |
| 70–79 | **0.57 [0.37–0.90]** | 0.99 [0.72–1.36] |
| >=80 | **0.58 [0.36–0.94]** | 0.93 [0.66–1.31] |
| Sex | | |
| Female | Reference | Reference |
| Male | 1.07 [0.98–1.18] | 0.97 [0.89–1.06] |
| Race | | |
| White | Reference | Reference |
| African Americans | **1.16 [1.02–1.33]** | 1.03 [0.90–1.18] |
| Others | 1.02 [0.87–1.20] | 1.10 [0.96–1.27] |
| Unknown | 0.35 [0.05–2.57] | 0.99 [0.29–3.37] |
| Location | | |
| Left | Reference | Reference |
| NOS | 0.97 [0.71–1.31] | 1.15 [0.90–1.48] |
| Rectum | 0.87 [0.75–1.01] | 0.95 [0.83–1.08] |
| Right | 1.05 [0.94–1.17] | **1.22 [1.10–1.35]** |
| Tumor size | | |
| <=2 cm | Reference | Reference |
| 2–4 cm | 0.95 [0.79–1.15] | 1.00 [0.85–1.19] |
| 4–6 cm | 1.00 [0.83–1.20] | 1.10 [0.94–1.29] |
| >6 cm | 1.06 [0.87–1.28] | **1.25 [1.05–1.48]** |
| N | 1.04 [0.86–1.27] | 1.01 [0.85–1.21] |
| Surgery | | |
| No | Reference | Reference |
| Yes | **0.77 [0.65–0.91]** | **0.86 [0.75–0.995]** |
| Stage | | |
| 0/(Tis) | Reference | Reference |
| I | 1.55 [0.22–10.88] | 1.07 [0.14–8.48] |
| II | 4.62 [0.69–30.73] | 2.96 [0.39–22.73] |
| III | **6.96 [1.06–45.70]** | 4.08 [0.54–30.82] |
| IV | **18.90 [2.88–123.93]** | **11.31 [1.50–85.33]** |
| Unknown | 5.52 [0.78–39.14] | 5.01 [0.64–39.46] |

| Table 3 (continued). | | |
| --- | --- | --- |
| **Characteristics** | **SH model** | |
| | **KRAS MT** **HR (95% CI)** | **KRAS WT** **HR (95% CI)** |
| Grade | | |
| Low grade (I & II) | Reference | Reference |
| High grade (III & IV) | **1.35 [1.20–1.52]** | **1.38 [1.25–1.53]** |
| NOS | 0.95 [0.81–1.10] | 1.07 [0.93–1.23] |
| Regional nodes positive | | |
| >=10 | Reference | Reference |
| 0 | **0.48 [0.39–0.59]** | **0.37 [0.31–0.46]** |
| 1–3 | **0.61 [0.52–0.73]** | **0.50 [0.43–0.57]** |
| 4–9 | **0.80 [0.68–0.95]** | **0.70 [0.61–0.80]** |
| Radiotherapy | | |
| No | | |
| Yes | 1.07 [0.92–1.26] | 1.00 [0.87–1.15] |
| Unknown | 1.04 [0.56–1.92] | 0.45 [0.19–1.09] |
| Chemotherapy | | |
| No | Reference | Reference |
| Yes | **0.77 [0.66–0.90]** | 0.99 [0.86–1.13] |
| Marital status | | |
| Married | Reference | Reference |
| Unmarried | **1.19 [1.08–1.32]** | **1.11 [1.01–1.21]** |
| Unknown | 0.88 [0.70–1.11] | 1.12 [0.91–1.37] |

Note:
KRAS MT, KRAS mutant; KRAS WT, KRAS wild type; HR, Hazard ratios; 95% CI, 95% confidence intervals; the widowed or single (never married or having a domestic partner) or divorced or separated patients was defined as unmarried; Tis, Tumor in situ; significant results were bolded.

$p < 0.05$) than KRAS WT patients. Further subgroup analysis found the KRAS mutation was an independent risk factor for the CSS of stage III (HR = 1.28 (95% CI [1.09–1.49]), $p < 0.05$) and stage IV (HR = 1.14 (95% CI [1.06–1.23]), $p < 0.05$) CRC patients. Moreover, we found KRAS shorten the CSS in patients with cancers occurred at left colon (HR = 1.28 (95% CI [1.15–1.42]), $p < 0.05$) and rectum (HR = 1.23 (95% CI [1.07–1.43]), $p < 0.05$) but not right colon (HR = 1.07 (95% CI [0.97–1.19]), $p > 0.05$).

## Multivariate SH analyses of each variable for the CSS of KRAS MT and KRAS WT CRC patients

As shown in Table 3, the multivariate SH model identified the absence of surgery, higher tumor stage and grade, and unmarried status as risk factors for both KRAS MT and KRAS WT CRC patients (HR > 1, $p < 0.05$). We observed there was no significant association between sex and the prognosis in neither KRAS MT nor KRAS WT CRC patients ($p > 0.05$).

Prognostic discrepancies were found in other variables between KRAS MT and WT CRC patients. The older age was a protective factor for KRAS MT patients (HR < 1,

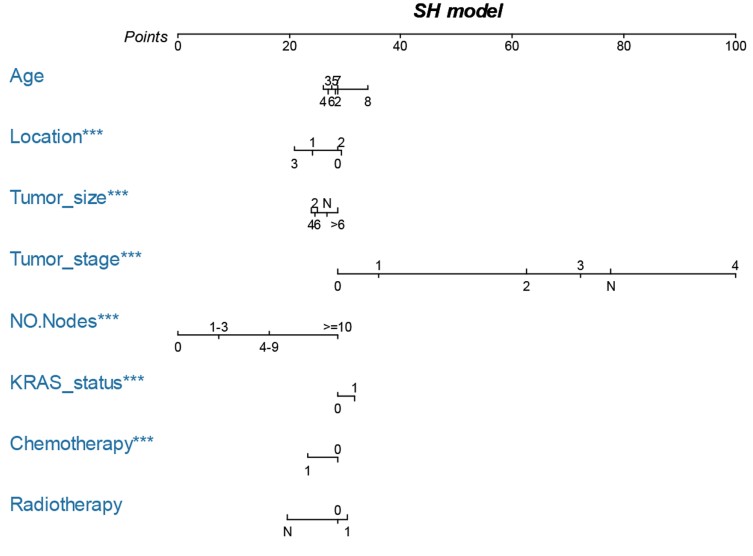

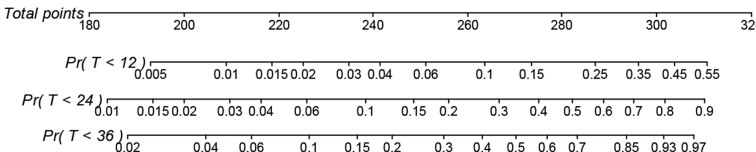

**Figure 3 Nomogram for predicting 1-year, 2-year and 3-year CSS of CRC patients who had KRAS testing.** The nomogram is used by summing the points identified on the top scale for each independent variable and drawing a vertical line from the total points scale to the 1-year, 2-year and 3-year CSS to obtain the probability of survival. The total points projected to the bottom scale indicate the % probability of the 3-year survival. Age: 2, 20–29 years, 3, 30–39 years, 4, 40–49 years, 5, 50–59 years, 6, 60–69 years and 7, 70–79 years; Race: 1, Caucasian, 2, African American, 3, Other race and N, Unknown race; Tumor size: 2, "0–2 cm", 4, "2–4 cm", 6, "4–6 cm", >6 = ">6 cm", N, Unknown size; Tumor stage, 0, 0 stage (Tumor in situ), 1, I stage, 2, II stage, 3, III stage, 4, IV stage and N, Unknown stage; No. Nodes, the number of positive regional lymph nodes; KRAS status: 0, Wild type and 1, Mutation; Chemotherapy, 0, none/unknown and 1, yes; Radiotherapy, 0, none/unknown or refused, 1, beam radiation or combination of beam with implants or isotopes or radiation with method or source not specified or radioactive implants or radioisotopes and N, Recommended, unknown if administered.

$p < 0.05$) but was not associated with the prognosis of KRAS WT patients ($p > 0.05$). We found that the race of African American was a risk factor for KRAS MT patients but not for KRAS WT patients. The right side colon cancer was observed to have worse CSS than left side colon cancer in KRAS WT patients (HR > 1, $p < 0.05$) but not in KRAS MT patients ($p > 0.05$). Moreover, we found that the chemotherapy was only a protective factor for KRAS MT patients (HR < 1, $p < 0.05$) but not for KRAS WT patients ($p > 0.05$).

## Nomogram construction and validation

The LASSO, SCAD and MCP analyses all selected age, location, tumor size and stage, regional positive nodes amount, KRAS mutation status, chemotherapy experience and radiotherapy experience as the key prognostic variables of our nomogram (Table 4). These variables were then used to construct the multivariate SH model based nomogram to

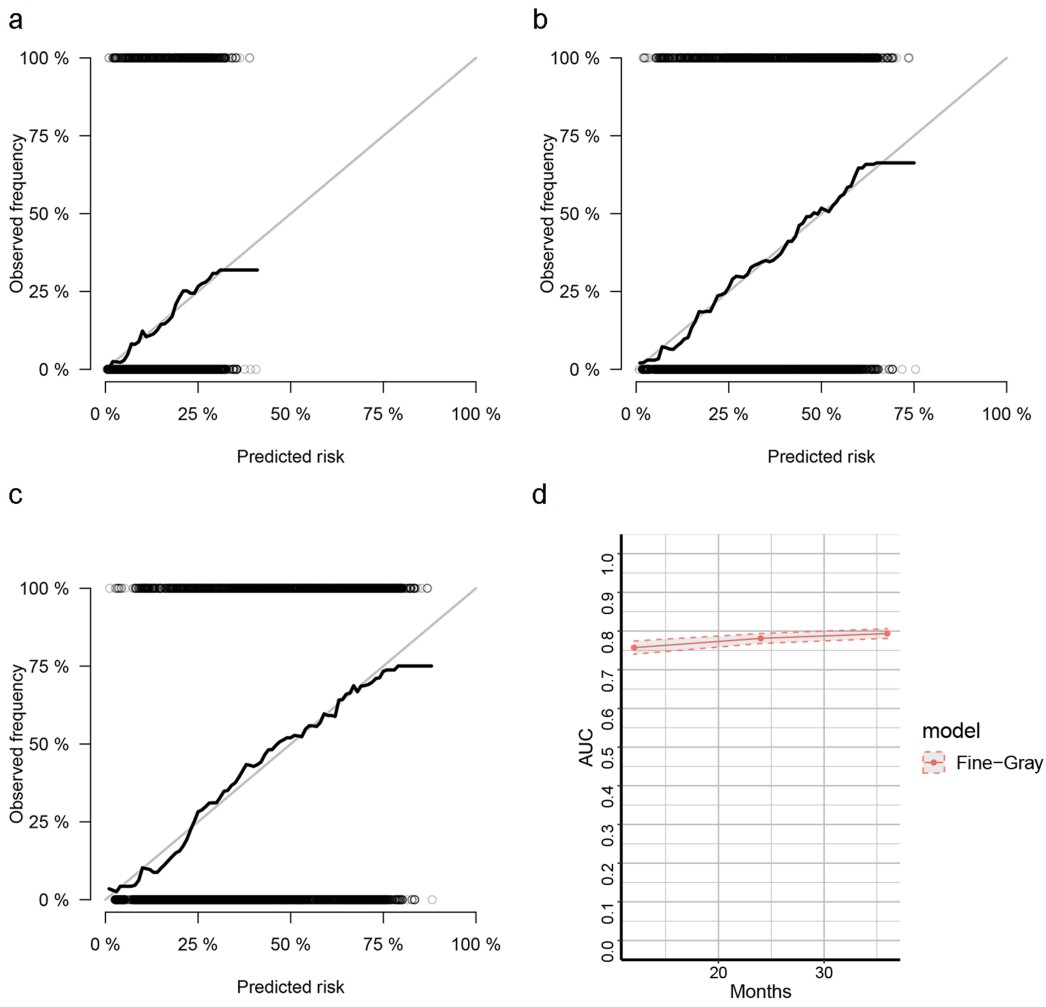

**Figure 4 Calibration curves for cox-based and SH based nomograms.** (A–C) The calibration plots for predicting 1-year, 2-year and 3-year CSS of CRC patients; (D) the AUC plots for SH-based nomogram.

predict the 1-year, 2-year and 3-year CRC specific death (Fig. 3). Internal validation showed good calibration (Figs. 4A–4C, there were good agreements between the nomogram predicted CRC death and actual observed CRC death) and discrimination (AUC > 0.75, Fig. 4D) of current nomogram.

## DISCUSSION

The KRAS testing for metastatic CRC patients was recommended by the National Comprehensive Cancer Network (NCCN). The rate of KRAS testing for metastatic or non-metastatic CRC patients was increased in recent years according to SEER database (Fig. S2). However, the association between KRAS mutation status and the prognosis of CRC patients remains unclear. A SEER based study (*Charlton et al., 2017*) found that there was no association between KRAS mutation status and the OS of CRC. This might be a result of the limited follow up time, as they included the 2010–2012 data meanwhile had a last follow-up time of December 2013. Compared with this study, we included with

**Table 4 Variable selection: estimated coefficients (SEs) for the current SH model.**

| Characteristics | LASSO | SCAD | MCP |
|---|---|---|---|
| Age | −0.022 | −0.025 | −0.028 |
| Sex | 0.000 | 0.000 | 0.000 |
| Race | 0.000 | 0.000 | 0.000 |
| Location | −0.032 | −0.041 | −0.044 |
| Surgery | −0.069 | 0.000 | 0.000 |
| Tumor size | 0.017 | 0.027 | 0.025 |
| Tumor stage | 0.226 | 0.233 | 0.233 |
| Grade | 0.000 | 0.000 | 0.000 |
| Regional nodes positive | 0.097 | 0.105 | 0.105 |
| KRAS status | 0.13 | 0.182 | 0.183 |
| Chemotherapy | 0.314 | 0.384 | 0.383 |
| Radiotherapy | −0.061 | −0.095 | −0.095 |
| Marital status | 0.000 | 0.000 | 0.000 |

Note:
LASSO, least absolute shrinkage and selection operator; SCAD, smoothly clipped absolute deviation (SCAD); MCP, measure–correlate-predict (MCP).

2010–2012 data while the last follow-up time was December 2015. The median survival time of our cohort was 33 months and the overall death rate of current study was 62.1%, indicating that our follow up time was relative sufficient. Furthermore, CRC patients were often diagnosed at an old age, therefore, competing risk analysis was more appropriate in the SEER based study. Our competing risk model found KRAS MT would shorten the CSS in CRC patients. Further subgroup analysis found that KRAS MT patients had worse CSS than KRAS WT patients among stage III or stage IV CRC patients or patients with left side colon cancer or rectal cancer. Moreover, the current study firstly built a competing nomogram for CRC patients who had KRAS testing.

Age was observed as a risk factor for the OS of CRC patients (*Charlton et al., 2017*; *Van Eeghen et al., 2015*). However, CRC patients are usually elders who might have high potential risk of deaths from other diseases. Our competing risk model found the older age was not associated with worse CSS of CRC patients. Moreover, older KRAS MT patients might have better CSS than young patients.

Left colon cancer was found to be more sensitive to anti-EGFR targeted therapy than right colon cancer (*Venook et al., 2017*). The right side colon cancer was found to have more BRAF mutation than left side colon cancer, which might cause the resistant to anti-EGFR therapy (*Van Brummelen et al., 2017*) and worsen the prognosis (*Salem et al., 2017*). Hence, for left-sided colon cancer, KRAS WT CRC patients are more likely to be benefit from anti-EGFR targeted therapy and have better outcomes than KRAS MT patients. Indeed, we found KRAS mutation was an independent risk factor for left side colon cancer but not right side colon cancer. Moreover, in KRAS WT patients, we found right colon cancer had worse CSS than left side colon cancer meanwhile in KRAS MT patients, there was no significant prognostic difference between right and left side colon cancers.

We built an SH model-based nomogram to predict the probability of cancer specific death after a variable selection. Our nomogram was well validated. The predictors of current nomogram were easy to be obtained in clinical use. The increasing concern about competing risk had promoted researchers to develop competing risk nomograms for a groups of cancers (*Brockman et al., 2015*; *Kattan, Heller & Brennan, 2003*; *Kutikov et al., 2010*; *Shen, Sakamoto & Yang, 2016*; *Yang, Shen & Sakamoto, 2013*).

There were certain limitations in our study. First, prognostic differences were found between KRAS codon 12 and codon 13 mutations (*Imamura et al., 2012*). However, the detailed KRAS mutation pattern was not registered in SEER. The detailed anti-EGFR therapy and chemotherapy strategy were also missed. Second, other genetic variables, such as BRAF mutation and microsatellite instability (MSI), were also frequently occurred in CRC and associated with the prognosis of CRC (*Jung, Kim & Kim, 2016*; *Sanz-Garcia et al., 2017*). These data were also not available in SEER. Third, selection bias might exist in current study as we only included patients with complete information of included variables.

## CONCLUSION

This is the first population based competing risk study on the association between KRAS mutation status and the CRC prognosis. We found that KRAS mutation would worsen the CSS for patients with stage III and stage IV CRC, and for patients with cancers in the locations of left side colon and rectum. We constructed an SH based nomogram with good discrimination and calibration which might help the clinicians to predict the 1-year, 2-year and 3-year cancer specific death of CRC patients who had KRAS testing.

### Funding
This grant was supported by the National Natural Science Foundation of China (81372178; 81502386; 81772944; 81572715) and the High level health innovative talents program in Zhejiang and Natural Science Foundation of Zhejiang (LZ17H60003). The funders had no role in study design, data collection and analysis, decision to publish, or preparation of the manuscript.

### Grant Disclosures
The following grant information was disclosed by the authors:
National Natural Science Foundation of China: 81372178, 81502386, 81772944 and 81572715.
Natural Science Foundation of Zhejiang: LZ17H60003.

### Competing Interests
The authors declare that they have no competing interests.

## Author Contributions

- Dongjun Dai conceived and designed the experiments, performed the experiments, analyzed the data, prepared figures and/or tables, authored or reviewed drafts of the paper, and approved the final draft.
- Yanmei Wang performed the experiments, analyzed the data, prepared figures and/or tables, and approved the final draft.
- Liyuan Zhu performed the experiments, analyzed the data, prepared figures and/or tables, and approved the final draft.
- Hongchuan Jin conceived and designed the experiments, authored or reviewed drafts of the paper, and approved the final draft.
- Xian Wang conceived and designed the experiments, authored or reviewed drafts of the paper, and approved the final draft.

## Data Availability

The raw data from the SEER database is available in the Supplemental File.

## Supplemental Information

Supplemental information for this article can be found online at http://dx.doi.org/10.7717/peerj.9149#supplemental-information.

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
