# Peer review of "Prognostic value of KRAS mutation status in colorectal cancer patients: a population-based competing risk analysis"

_PeerJ, doi:10.7717/peerj.9149_

## Round 0.1 · original submission · Major Revisions

Please respond to the reviewers. More information on KRAS in the introduction and 95% confidence intervals on results, please.

Reviewer 1 ·

Basic reporting

The manuscript was written well and English is good.

Experimental design

I would like to ask IRB approved at hospital for this clinical study

Validity of the findings

this study seems to use a more specific statistical method and nomogram was made.
Please explain this more easiler to readers.

Additional comments

this study was done whether K ras mutaiton can impact the prognosis of CRC using large patiens cohort ( SEERS data). author analyze prognostic factor including K ras mutation in CRC, In addition to that, they also want to develop nomogram to predict prognosis. But data itself showed too hetergenous and detailed information about chemotherapy regimen and radiotherapy in rectal cancer, as well known, there have been many prognositc factos TNM etc. but only K ras mutation was compared with other clinicopathologic factors, Author also indicated the limitation of this stduy it need a other genetic mutation such as BRAF and MSI etc . Regading nomogram and many figures did not geve us confidence,

Reviewer 2 ·

Basic reporting

Written in a language foreign to the authors
There are a good number of grammatical and linguistic errors.
The introduction and discussion do not provide sufficient information about the known status of KRAS mutation and prognosis form colorectal cancer
There is another paper by Charlton et al
J Natl Compr Canc Netw. 2017 Dec;15(12):1484-1493. doi: 10.6004/jnccn.2017.7011.
KRAS Testing, Tumor Location, and Survival in Patients With Stage IV Colorectal Cancer: SEER 2010-2013.
Charlton ME1, Kahl AR1, Greenbaum AA2, Karlitz JJ3, Lin C4, Lynch CF1, Chen VW5

This study has used the same data and included a similar number of patient and reported largely similar findings. The differences between the current study and the Charlton study should be clearly detailed.

Experimental design

This is a statistical study and it would be appropriate to get a statistics expert to review the statistical methods used.

Validity of the findings

If the statistical methods are appropriate the results appear valid and will be of some clinical value

Additional comments

There is some novel data about the effect of KRAS mutation status in Stage III cancer patients.
Written in a language foreign to the authors
There are a good number of grammatical and linguistic errors.
The introduction and discussion do not provide sufficient information about the known status of KRAS mutation and prognosis form colorectal cancer
Re the paper by Charlton et al
J Natl Compr Canc Netw. 2017 Dec;15(12):1484-1493. doi: 10.6004/jnccn.2017.7011.
KRAS Testing, Tumor Location, and Survival in Patients With Stage IV Colorectal Cancer: SEER 2010-2013.
Charlton ME1, Kahl AR1, Greenbaum AA2, Karlitz JJ3, Lin C4, Lynch CF1, Chen VW5

This study has used the same data and included a similar number of patient and reported largely similar findings. The similarities and differences between the current study and the Charlton study should be clearly detailed.

---

## Round 0.2 · accepted · Accept

Please address any minor grammatical issues at the proof stage.

Reviewer 2 ·

Basic reporting

Much improved
There remain some grammatical errors that could be corrected by the editorial team

Experimental design

They have responded satisfactorily to the reviewers comments

Validity of the findings

Better presented

Additional comments

Minor grammatical errors remain